# Significance of Interleukin (IL)-4 and IL-13 in Inflammatory Arthritis

**DOI:** 10.3390/cells10113000

**Published:** 2021-11-03

**Authors:** Milena Iwaszko, Sylwia Biały, Katarzyna Bogunia-Kubik

**Affiliations:** Laboratory of Clinical Immunogenetics and Pharmacogenetics, Hirszfeld Institute of Immunology and Experimental Therapy, Polish Academy of Sciences, 53-144 Wrocław, Poland; sylwia.bialy@hirszfeld.pl (S.B.); katarzyna.bogunia-kubik@hirszfeld.pl (K.B.-K.)

**Keywords:** inflammatory arthritis, rheumatoid arthritis, psoriatic arthritis, ankylosing spondylitis, spondyloarthritis, IL-4, IL-13

## Abstract

Interleukin (IL)-4 and IL-13 belong to the T helper 2 (Th2) cytokine family, along with IL-3, IL-5, and IL-9. These cytokines are key mediators of allergic inflammation. They have important immunomodulatory activities and exert influence on a wide variety of immune cells, such as B cells, eosinophils, basophils, monocytes, fibroblasts, endothelial cells, airway epithelial cells, smooth muscle cells, and keratinocytes. Recent studies have implicated IL-4 and IL-13 in the development of various autoimmune diseases. Additionally, these cytokines have emerged as potential players in pathogenesis of inflammatory arthritis. Recent findings suggest that the IL-4 and IL-13 might play a significant role in the downregulation of inflammatory processes underlying RA pathology, and beneficially modulate the course of the disease. This review summarizes the biological features of the IL-4 and IL-13 and provides current knowledge regarding the role of these cytokines in inflammatory arthritis.

## 1. Introduction

Inflammatory arthritis constitutes a group of various rheumatic diseases characterized by an inflammation of synovial joints as well as systemic manifestations. Rheumatoid arthritis (RA), psoriatic arthritis (PsA) and ankylosing spondylitis (AS) are the most widespread subtypes of inflammatory arthritis. The worldwide prevalence of inflammatory arthritis amounts to approximately 3%. The exact etiology of the RA, PsA, and AS remains largely unknown; however, an imbalanced cytokine network plays a key role in the pathogenesis of these conditions.

Interleukin (IL)-4 and IL-13 are members of the T helper (Th) 2 family of cytokines. This group also includes IL-3, IL-5, and IL-9. Genes encoding the IL-4 and IL-13 in humans are located on chromosome 5q31 within a cluster of Th2-related cytokine genes including *IL-3*, *IL-5*, and *IL-9* [1]. IL-13 was first described in 1983 as a protein secreted by activated mouse Th2 cells, and a human IL-13 was cloned in 1993 [2,3,4]. IL-13 represents a four-helix-bundle cytokine with two disulfide bridges [5,6]. IL-13 has a single open-reading frame of 132 amino acids with a 20-amino-acid sequence that is cleaved and secreted as a 10-kd unglycosylated mature protein [7]. The gene encoding IL-13 has four exons and three introns and is located 12 kb upstream of the IL-4 encoding gene [8]. IL-4 was discovered in 1982 as a factor with the capacity to induce proliferation of murine B lymphocytes [9]. Cloning of a gene encoding both mouse and human IL-4 was performed in 1986 by Noma et al. and Yokota et al. [10,11]. IL-4 and IL-13 share 25% similarity at amino acid level and have overlapping functions [12]. The IL-4 structure is also based on a four-helix bundle and contains three disulfide bridges [13]. The human IL-4’s cDNA encodes 153 amino acids protein with a 129-amino-acid sequence which is cleaved from the pre-protein and secreted as 15.4-kD unglycosylated mature protein [12]. The gene encoding IL-4 comprises four exons and three introns. Th2 cells are the main source of IL-4 and IL-13; however, lower levels of these cytokines might be also secreted by a wide range of other cells, including NK cells, Th1 cells, CD8+ T cells, innate lymphoid type 2 cells, B lymphocytes, mast cells, macrophages, basophils, and eosinophils [12,14,15,16,17,18].

IL-4 and IL-13 play a critical role in allergic inflammation and parasite infection [19,20,21,22,23]. These cytokines have the capacity for switching immunoglobulin (Ig) class of IgE and IgG4 [24,25,26]. They stimulate B cell proliferation, and activation of eosinophils, basophils, and mast cells [27,28,29,30]. Additionally, they are involved in collagen production by fibroblasts, and they induce vascular cell adhesion molecule (VCAM)-1 expression on endothelial cells [31]. Furthermore, IL-4, although not IL-13, acts a key role in promotion of Th2 differentiation [16,17,32,33]. On the other hand, IL-13 triggers mucus production, goblet cells metaplasia and proliferation, as well as contraction of smooth muscle cells resulting in airways remodeling [34,35].

These cytokines display the capacity to antagonize Th1-driven proinflammatory immune response. They downregulate synthesis of many proinflammatory cytokines, including IL-1, IL-6, IL-10, IL-12, TNF-α, and macrophage migration inhibitory factor (MIF) [36]. They are also involved in inhibition of proinflammatory chemokines, such as IL-8, monocyte chemotactic protein-3 (MCP-3), interferon-gamma inducible protein 10 kD (IP-10), or macrophage inflammatory protein-3 (MIP-3) [37]. They also suppress other mediators of inflammation, such as granulocyte-macrophage colony-stimulating factor (GM-CSF), prostaglandins, metalloproteinases, reactive oxygen, and nitrogen intermediates [38,39]. They are also key players in the process of macrophage polarization towards anti-inflammatory profile [40]. In addition, these cytokines inhibit Th17 development and suppress release of cytokines by Th17 cells resulting in downregulation of Th17-mediated inflammation [41,42]. On the other hand, they upregulate an interleukin-1 receptor antagonist (IL-1Ra) and an interleukin 1 receptor type II (IL-1RII) that are involved in suppression of inflammation [43,44]. Anti-inflammatory roles of IL-4 and IL-13 are well documented in the pathogenesis of psoriasis, type I diabetes, and experimental autoimmune encephalomyelitis (EAE)—the animal model of multiple sclerosis [45,46,47]. Additionally, these cytokines have emerged as potential players in pathogenesis of inflammatory arthritis [48].

IL-4 and IL-13 share one receptor, designated as a type II receptor. This receptor is a heterodimer consisting of an IL-4 receptor (IL-4R) α and an IL-13 receptor (IL-13R) α1 subunit (Figure 1) [36,49]. The IL-4Rα subunit is also a component of type I receptor that interacts only with IL-4 [50,51,52]. IL-13 can also bind another receptor, namely IL-13Rα2, which is regarded as a decoy receptor [36]. An interaction of IL-13 or IL-4 with the type II receptor activates a tyrosine kinase proteins Janus kinase 1 (JAK1) and a tyrosine kinase 2 (TYK2), resulting in phosphorylation of a signal transducer and activator of transcription (STAT) 6 and its translocation to a nucleus [53,54,55]. This interaction might also lead to activation of other signaling pathways, such as a STAT3, a phosphatidylinositol 3-kinase (PI3K), and a mitogen activated protein kinase (MAPK). The type II receptor complex is expressed on a wide range of cells, such as B cells, eosinophils, basophils, monocytes fibroblasts, endothelial cells, airway epithelial cells, smooth muscle cells, and keratinocytes [56].

## 2. Regulation of *IL-4*/*IL-13* Genes Expression

Th2 cytokines production is governed by a network of factors comprising T-cell receptor (TCR) stimulation with antigenic peptide/main histocompatibility complex class II (MHC-II), with antigen dose playing a key role, co-stimulatory molecules, and an appropriate cytokine environment [57]. The canonical pathway for IL-4/IL-13 production is activated by IL-4 leading to the phosphorylation of STAT6 and an upregulation of GATA binding protein 3 (GATA3) [58]. The interaction between TCR and a peptide/MHC-II complex presented by an antigen-presenting cell (APC) is also involved in the activation of GATA3 expression. GATA3 constitutes the master of Th2 cell transcription regulation. However, IL-4/IL-13 production could be induced by non-canonical signaling pathways independently of IL-4 action. The IL-2-mediated STAT5 pathway is also a crucial element promoting production of Th2 cytokines. Notch signaling is another non-canonical pathway involved in IL-4/IL-13 expression upregulation. In addition, stimulation of TCR also activates a nuclear factor of activated T cells (NFAT) and activator protein-1 (AP-1), which are involved in *IL-4*/*IL-13* transcriptional regulation [59]. NFAT induction is mediated by calcium-calcineurin signaling, and the AP-1 is triggered via protein kinase C (PKC)/Ras-dependent pathway. AP-1 transcription factors comprise members of the Jun and Fos protein families characterized by a leucine zipper domain. NFAT binds to the *IL-4* and *IL-13* promoter region and cooperates with AP-1 family members (Fos/Jun), forming a transcriptionally active complex. An important role in the regulation of *IL-4* transcription has also been attributed to a c-Maf, basic region/leucine zipper transcription factor. Additionally, several other transcription factors are involved in orchestrating *IL-4*/*IL-13* gene expression, including nuclear factor-κB (NF-κB), nuclear factor of IL-6 (NF-IL6), transcription factor CP2 (TFCP2), nuclear transcription factor Y (NFY), and Yin-yang 1 (YY1) [60].

## 3. IL-4 and IL-13 Polymorphism in Inflammatory Arthritis

One of the commonly examined versions of *IL-13* is a single nucleotide polymorphism rs1800925 (-1055 C/T; promoter region). *IL-13* rs1800925 has been studied in RA patients; however, no associations have been found between this genetic variant and a susceptibility to the disease in three different studies comprising Chinese and Caucasian populations [61,62,63,64]. Although stratification analyses in the study by Wang et al. revealed that the C allele was associated with RA risk in Chinese patients with erythrocyte sedimentation rate (ESR) < 25.00 [64].

*IL-13* polymorphisms, such as rs1800925, rs20541 (G/A; Arg130Gln; exon 4), and rs848 (C/A; 3′ untranslated region), have also been studied in the context of PsA and psoriasis. Duffin et al. reported that among psoriasis patients the rs1800925 T, rs20541 A, and rs848 A alleles have the effect of protection from PsA [61]. In line with these findings, significant associations between the rs1800925 C, rs20541 G, and rs848 C polymorphisms and PsA susceptibility have been found in two other studies conducted on Caucasian population [65,66]. The rs848 C genetic variant and the rs1800925 C allele were also found to be associated with increased risk of PsA development in psoriasis patients in the study by Eder et al. [65].

The most commonly studied *IL-4* polymorphism in RA was rs2243250 (-590 C/T; promoter region). The *IL-4* -590 T allele was found to be associated with RA susceptibility in Spanish and Chinese populations [67,68]. In the study by Pawlik et al., the presence of the T allele of rs2243250 was correlated with the active form of RA and increased parameters of disease activity, including Disease Activity Score 28 (DAS28), ESR, and number of swollen and tender joints. However, an association between rs2243250 and RA predisposition was not observed [69]. In line with these findings, another study conducted by Hussein et al. reported an association of the TT genotype of rs2243250 with erosive RA, presence of anti-cyclic citrullinated peptides (anti-CCP), and increased disease severity parameters. In addition, authors observed that the TT genotype was associated with susceptibility to RA [70]. On the contrary, three other studies did not find significant relationships between the rs2243250 polymorphism and RA risk [62,71,72]. However, the results from two meta-analyses confirmed that the T allele of the rs2243250 genetic variant was significantly associated with increased risk of RA development [73,74].

In a multicohort candidate gene study investigating 8 SNPs within the *IL-4* gene, no associations were found in regard to joint damage in the course of RA [75]. Some studies examined also rs79071878, *IL-4*’s variable number of tandem repeats (VNTR; 70 bp; 3 intron). Genevay at al. reported that the presence of the rare *IL-4* VNTR (2) allele positively correlated with lower joint destruction in RA patients [76]. The protective effect of the rare *IL-4* VNTR (2) allele was also observed with regard to joint destruction [77]. Conversely, one study reported association of the *IL-4* VNTR (2) allele with RA risk [71]. Although no significant association was detected between the *IL-4* VNTR polymorphism and RA susceptibility, nor with rheumatoid factor (RF) presence and disease severity [68].

Several genetic studies also investigated polymorphisms within the *IL-4R* gene: rs1805010 (+148 A/G; I50V; exon 5) and rs1801275 (+1902 A/G; Q551R; exon 12). They have important functional significance. *IL-4R* I50V polymorphism is located in the extracellular domain and affects strength of interaction with IL-4, while *IL-4R* Q551R variant is located in the intracellular domain and affects intracellular signaling. In the study by Prots et al. on rs1805010, the GG genotype (V50/V50) was found to be associated with bone erosion in RA patients [78]. However, no relationship between the *IL-4R* I50V polymorphism and RA development was detected [78]. On the other hand, Marinou et al. did not find any association of this genetic variants with RA risk or disease severity [62]. Discordant results were reported by Moreno et al., with the *IL-4R* A allele (I50) being correlated with a presence of RF and a history of articular joint replacement [68]. Meta-analysis performed by Peng et al. revealed that the G allele (V50) of the *IL-4R* rs1805010 variant might be a risk factor for RA [79]. Interesting results were obtained in the study by Wallis et al. investigating influence of the *IL-4R* I50V polymorphism on Th17 production in cell culture experiments [80]. The authors observed that IL-4 effectively inhibited IL-17 secretion by cells obtained from patients with the *IL-4R* AA (I50/I50) or AG (I50/V50) genotypes, but it failed to suppress production of IL-17 in the case of the GG genotype (V50/V50) [80].

No association was found between *IL-4R* Q551R polymorphism and disease susceptibility or radiographic progression in RA patients [68]. In line with these findings, the Q551R polymorphism was not associated with RA risk, nor joint damage progression, in three other studies [62,76,77]. However, in another study a significant relationship between the Q551R G allele (R551) genetic variant and RF in RA patients was detected [78]. No association was detected between the Q551R polymorphism and RA risk in meta-analysis performed by Peng et al. [79]. Both the Q551R and I50V polymorphisms were also investigated in PsA patients. However, no significant associations were found with regard to PsA susceptibility or disease severity [81].

In summary, the *IL-13* rs1800925 C allele was associated with increased PsA risk; however, no association with RA susceptibility was detected. Additionally, the *IL-13* rs20541 G and rs848 C alleles were found to correlate with PsA risk. The results regarding *IL-4* rs2243250 are contradictory. However, most of the performed studies, including two meta-analyses, reported association of the *IL-4* rs2243250 T allele with RA risk, as well as worse course of disease. On the other hand, a protective effect of the *IL-4* VNTR (2) allele was found in context of RA, but two other studies did not confirm these results. The results regarding the rs1805010 polymorphism within the gene encoding *IL-4R* are not consistent, with some studies indicating a protective, and some a deleterious effect of the G allele on RA development. The second-most commonly studied *IL-4R* polymorphism, rs1801275, was not found to be associated with RA and PsA risk nor with RA disease course.

## 4. IL-4 and IL-13 Serum Concentrations in Inflammatory Arthritis

Isomaki et al. observed that IL-13 is consistently present in a synovial fluid (SF) of RA patients [82]. In the study by Spadaro et al., serum levels of IL-13 were significantly higher in RA patients than in healthy individuals [83]. However, the IL-13 serum concentrations did not differ between PsA patients and controls. On the other hand, increased levels of IL-13 in SF have been found in both RA and PsA patients, as compared to osteoarthritis control group [83]. Upregulation of the IL-13 levels in serum was also observed in patients diagnosed with PsA in the study by Szodoray et al. [84].

In addition, increased concentrations of IL-13 were observed in sera of early RA patients, as compared to healthy controls, in the study by Silosi et al. In this study, a positive correlation between the IL-13 serum levels and disease activity was also found [85]. Higher IL-13 levels in serum of RA patients were also detected in other studies [86]. In the study conducted by Tokayer et al. increased concentrations of IL-13 were observed in both serum and SF of RA patients [87]. Additionally, elevated levels of IL-13 were found in SF of patients diagnosed with early RA [88]. However, contradictory results were observed in the study by Azizieh et al. In this study, IL-13 levels did not differ from that of control group and no correlation was observed between IL-13 concentrations and disease activity [89]. On the other hand, two research groups reported decreased IL-13 levels in both serum and SF of RA patients [90,91].

IL-4 is expressed at relatively low levels in peripheral blood (PB) and SF [92,93,94]. Increased mRNA levels of IL-4 in the whole blood of RA patients were reported in the study Kawashima et al. [95]. At a protein level, IL-4 was found to be increased in mononuclear and whole blood cells from RA patients in response to in vitro stimulation [96,97]. Enhanced IL-4 concentrations were also found in RA SF and plasma samples [98,99]. In addition, Kokkonen et al. observed increased IL-4 levels in RA patients before disease development [100]. Conversely, Constantin et al. reported downregulated expression of the *IL-4* gene in mononuclear cells obtained from RA patients as compared to healthy individuals [92]. They also demonstrated negative correlation between *IL-4* gene expression and clinical parameters of the disease [92]. In addition, Kramer et al. also observed that circulating levels of IL-4 were inversely correlated with disease activity in RA patients [101]. On the other hand, IL-4 secretion by SF and PB CD4+ and CD8+ cells from patients with RA was investigated in the study by Isomaki et al. Expressions of IL-4 by CD4+ and CD8+ cells were comparable between PB and SF cells in this study [102]. However, other study found that higher levels of IL-4 were secreted by CD8+ T cells derived from SF than from CD8+ T PB cells of RA patients [103].

In conclusion, the results regarding the IL-4/IL-13 levels in patients diagnosed with inflammatory arthritis are somewhat contradictory. Some studies have reported increased serum and SF levels of IL-13 in patients diagnosed with RA or PsA. However, some studies have documented contrary findings. The results regarding IL-4 concentrations in patients with inflammatory arthritis are also not consistent. It is difficult to compare the results derived from these studies, since the IL-4/IL-13 levels were measured in different cell populations, conditions, and experimental protocols. One of the most important aspect that should be considered regarding these results is disease duration. Three of the aforementioned studies examined patients with early RA and reported significantly elevated levels of IL-4/IL-13 [85,88,100]. Notably, one of these studies documented enhanced IL-4/IL-13 levels present in patients with early inflammatory arthritis, but no longer detected in patients with established RA [88]. These results suggest that cytokines profile of early arthritis patients is transient and differs from cytokines profile characterizing patients with an established RA. In line with this hypothesis, the IL-4/IL-13 cytokines might exert a diverse influence depending on the disease status. It is conceivable that these cytokines might be involved in a pathogenesis of inflammatory arthritis in a time-dependent manner, i.e., having an activatory influence at the disease’s onset and an inhibitory effect when the disease is established.

## 5. IL-4 and IL-13 Effect on Macrophages Polarization

IL-4, along with IL-13, is a potent inducer of macrophage polarization into the M2 phenotype (Figure 2). The M1 term refers to “classically activated” proinflammatory macrophages [48,104,105]. This subset of macrophages is characterized by expression of MHC-II molecules, CD40, CD80 costimulatory molecules, CD44 antigen, and a mannose receptor CD206 [106]. M1 macrophages release large amounts of proinflammatory cytokines, including IL-1β, IL-6, IL-8, IL-12, and tumor necrosis factor (TNF). Moreover, they produce nitric oxide and proteolytic enzymes, such as matrix metalloproteinase (MMP)-1, MMP-3, and MMP-13 [107,108]. M1 macrophages display proinflammatory activity and contribute to inflammation within joints [109,110,111]. Conversely, M2 macrophages are “alternatively activated” by IL-13 and IL-4 cytokines [82,112,113,114]. They represent anti-inflammatory phenotype and have regulatory and suppressive functions [115,116,117,118,119]. Phenotypically M2 macrophages exhibit upregulated expression of a CD16 marker and a scavenger receptor CD163, as well as the IL-1Ra, and the IL-1RII receptor. M2 macrophages produce anti-inflammatory cytokines, such as IL-10, IL-1α, transforming growth factor β, and tissue repair factors, leading to resolution of inflammation and suppression of arthritis [120,121].

## 6. IL-13 Effect on TNF Production

IL-13’s capacity to inhibit lipopolysaccharide (LPS)-induced production of TNF and IL-1β cytokines was examined in sets of in vitro experiments in the study by Hart et al. [122]. The authors found that IL-13 inhibited TNF production by mononuclear cells from PB of patients diagnosed with chronic inflammatory arthritis. However, IL-13 did not mediate downregulation of TNF production in experiments with mononuclear cells derived from SF. An inverse association was observed in case of IL-1. IL-13 significantly reduced IL-1 secretion by mononuclear cells from SF but not PB. The authors also investigated a response to IL-13 by monocytes cultured with granulocyte-macrophage colony-stimulating factor (GM-CSF) or M-CSF, and they reported that IL-13 suppressed LPS-induced IL-1 production, but an impact on TNF secretion was not detected [122]. However, Isomaki et al. demonstrated that IL-13 significantly reduced production of both LPS-induced TNF and IL-1 by SF mononuclear cells derived from RA patients [82]. In addition, Morita et al. reported IL-13-mediated downregulation of TNF, IL-1, as well as IL-6, and IL-8, by cultured synovial tissue (ST) cells from patients with RA [123]. Additionally, suppression of TNF, IL-1, and stromelysin-1 secretion mediated by IL-13 was observed in the study by Jovanovic et al. [124].

## 7. Il-4 and IL-13 In Vitro and Ex Vivo Models of Arthritis

Radstake et al. investigated IL-13 involvement in RA pathogenesis in the context of its potential role in dendritic cells’ (DCs) regulation via modulation of immunoglobulin Fc γ receptor (FcγR) expression [125]. They observed IL-13-mediated upregulation of inhibitory FcγRIIb receptor expression on cultured monocyte-derived DCs from healthy donors resulting in inhibitory DC phenotype and reduced production of inflammatory cytokines. However, RA-derived DCs were unresponsive to IL-13 and this effect was not observed in case of cells obtained from RA patients. These results suggest disruption of IL-13-mediated regulation of the inhibitory FcγRIIb receptor on DCs in RA patients that might lead to its aberrant expression [125].

In addition, the antiapoptotic effect of IL-13 on synovial cells was demonstrated by Relic et al. The authors observed that IL-13 protected both synoviocytes as well as cultured synovial explants from RA patients from apoptosis [126]. IL-13 was also found to upregulate expression of vascular cell adhesion molecule 1 (VCAM-1) in cultured human osteoblasts [127].

Chabaud et al. studied an influence of various cytokines, including IL-13, on MIP-3 in in vitro experiments with ST explants from RA patients. The MIP-3 is an important protein involved in inflammation and is produced by RA synoviocytes. The authors reported that IL-13 significantly reduced secretion of the MIP-3 by RA synoviocytes [128].

In an ex vivo model of RA synovitis with explants of synovium from RA patients, a significant reduction of proinflammatory cytokines, such as IL-1β, IL-6, and TNF, was observed in the presence of IL-4 [129]. Furthermore, in the same ex vivo model of synovitis, Dechanet et al. reported that IL-4 effectively inhibited proliferation of arthritis synoviocytes induced by growth factors, such as platelet-derived growth factor, and IL-1 [130]. In line with these findings, Morita et al. observed that IL-4 inhibited both secretion of proinflammatory cytokines, such as IL-1β, TNF, IL-6, and IL-8, as well as proliferation of fibroblast-like synoviocytes (FLS) obtained from synovial tissue of RA patients [123]. In the study by Woods et al., the influence of *IL-4* adenoviral gene therapy on RA synovium ex vivo was examined. They observed diminished levels of IL-1β, TNF, IL-8, and prostaglandin E2 (PGE2) in RA ST explants upon the adenovirally delivered IL-4 [131]. In addition, in a recently performed study with K/BxN serum transfer model and a set of in vitro experiments, it was shown that IL-4 inhibits proinflammatory neutrophils activity and their migration to inflamed joints [132]. It was also observed that IL-4 displays the capacity to inhibit secretion of IL-11 proinflammatory cytokine by cultured synovial cells obtained from RA patients [133]. On the other hand, it was observed in an in vitro study that IL-4 has an antiapoptotic effect on synovial cells [126].

It was also documented that IL-4 suppresses IL-1-induced production of MMP-1 and MMP-3 by cultured FLS [134]. IL-4 was also found to suppress vascular endothelial growth factor (VEGF) production by FLS from RA patients displaying anti-angiogenic effect [135]. Hart et al., in a set of in vitro experiments with LPS-stimulated monocytes, observed that IL-4 administration resulted in blockade of IL-1, TNF, and PGE2 production [136]. Furthermore, two other studies reported that IL-4 inhibits production of IL-6 by both monocytes isolated from RA blood and synovium [137,138].

Additionally, the effect of IL-4 on a bone resorption was assessed using ex vivo model of bone resorption. IL-4 was found to inhibit osteoclast activation and survival, as well as to suppress bone-resorbing cytokines, in juxta-articular samples of bone obtained during joint surgery [139]. Additionally, the in vitro protective effect of IL-4 against human cartilage degradation induced by antigen-stimulated mononuclear cells was documented in the study by Roon et al. [140]. A negative correlation between *IL-4* gene expression and the clinical activity of the disease was also observed in the PB mononuclear cells, as well as SF mononuclear cells, from RA patients [92,141].

In a recent study by Steel et al., an application of a fusion protein, consisting of IL-4 and IL-10, was tested in RA. The beneficial effects of the IL4-10 fusion protein obtained in this study comprised: in vitro inhibition of pro-inflammatory cytokines secretion, such as IL-1β, TNF-α, IL-6, and IL-8, by stimulated whole blood cells; ex vivo suppression of proinflammatory cytokines production by ST of arthritic patients; as well as diminished disease severity in proteoglycan-induced arthritis (PGIA) mouse model [142].

In conclusion, IL-4/IL-13 cytokines are involved in a reduction of production of proinflammatory cytokines (IL-1β, IL-6, IL-8, TNF) and metalloproteinases. They also inhibit the activatory FcγR and the inflammatory MIP-3 proteins expression, as well as they suppress neutrophils activity and protect synoviocytes from apoptosis. The summary of IL-4/IL-13 action is presented in Figure 3.

## 8. Effect of Methotrexate on IL-4

Several studies have investigated an impact of a methotrexate (mtx) treatment on IL-4 production in RA. However, the obtained results are not consistent. Constantino et al. found that the mtx exerts an in vitro effect on *IL-4* gene expression. The authors demonstrated that *IL-4* gene expression was significantly increased by stimulated mononuclear cells from RA patients in presence of the mtx and a phytohemagglutinin (PHA), which suggests that IL-4 constitutes the mtx therapeutic target in RA [92]. Conversely, in the study by Kremer et al., no difference was observed with respect to IL-4 serum levels in a course of the mtx treatment. However, inverse correlation of IL-4 serum concentrations and disease activity was detected in this study [143].

## 9. IL-13 Role in Bone Metabolism

It has been documented that IL-13 plays a key role in bone metabolism and inhibits bone resorption. In the study conducted by Onoe et al., it was observed that IL-13 displays the capacity to suppress cyclooxygenase-2-dependent prostaglandin biosynthesis in osteoblasts [144]. It has been also found that IL-13 inhibits osteoclastogenesis by decreasing expression of a receptor activator of nuclear factor κ B (RANK) and RANK ligand (RANKL) (Figure 3) [145]. In addition, IL-13-mediated suppressive effect on the osteoclastogenesis is also associated with its modulation of expression of an osteoprotegerin (OPG), which is a potent osteoclastogenesis inhibition factor [145,146].

A set of experiments with cultured mouse calvarial bones and bone marrow-derived macrophage (BMM) culture systems has shown that IL-13 effectively induces osteoblasts to the OPG production [145,146]. Similar results were also obtained in the study investigating effect of IL-13 on human umbilical vein endothelial cells [147]. In this study, IL-13 induced OPG expression in endothelial cells on both mRNA and protein levels [147]. An inhibitory effect of IL-13 has been also documented in vivo with a murine air pouch model [148]. Administration of IL-13 into the air pouch of mice resulted in an inhibition of an osteoclast differentiation as well as induction of the OPG production by osteoblasts [148].

In summary, IL-4/IL-13 cytokines downregulate the expressions of the RANK, RANKL, and PGE2, as well as enhancing the production of OPG, leading to the suppression of osteoclastogenesis and bone resorption.

## 10. IL-4 and IL-13 in Experimental Models of RA

Woods et al., using tissue explant model of RA, showed that *IL-13* gene therapy decreases secretion of proinflammatory cytokines [131]. IL-13’s role was studied in a mouse model of RA—collagen-induced arthritis (CIA). Administration of IL-13, using vector cells engineered to produce this cytokine, resulted in attenuation of the disease significantly reducing joint inflammation [149]. In further experiments, Bessis and co-workers treated TNF-α-transgenic mice with Chinese hamster ovary (CHO) cells engineered to produce IL-13. In this animal model of chronic inflammation, they demonstrated a significant inhibition of proinflammatory cytokines expression following IL-13 administration via gene therapy [150]. These observations were also confirmed in a gene therapy approach using xenogeneic cells engineered to secrete IL-13 and encapsulated into permeable hollow fibers. In a murine model of CIA, IL-13 secreted by such engineered encapsulated cells resulted in significant suppression of joint inflammation [151]. It has been also observed that IL-13 is necessary for the IL-25-mediated suppression of CIA. A protective effect of IL-25 administration in CIA mice was not observed after blocking IL-13 with monoclonal antibodies [152].

In another well-established model of RA, rat adjuvant-induced arthritis (AIA), Woods et al. studied the effect of *IL-13* gene therapy administered as intra-articular injections of adenoviruses bearing the *IL-13* gene. The authors observed that IL-13 delivered by gene therapy reduced inflammation, vascularization, level of proinflammatory cytokines, and bone destruction in arthritic joints, leading to significant amelioration of the disease. In addition, this beneficial effect was observed regardless of *IL-13* gene therapy being administered before or after arthritis onset [153]. Further experiments, conducted by this research group in the rat model of RA, revealed inhibition of angiogenesis in response to *IL-13* gene delivery in arthritic joints [154]. In another mouse model, immune-complex-mediated arthritis (ICA), an impact of *IL-13* gene therapy on cartilage matrix destruction was examined. It has been shown that IL-13 significantly reduced chondrocyte death and MMPs mediated cartilage matrix degradation [155].

Very interesting results were obtained in the recent study by Chen et al. regarding the influence of *Nippostrongylus brasiliensis* infection on arthritis in two different mouse models: serum-induced arthritis (SIA) and human TNF transgenic model (hTNFtg) [156]. The helminths infection led to an activation of Th2 responses and resulted in arthritis inhibition. Moreover, it was observed that the IL-4/IL-13/STAT6 pathway was critical for the *N. brasiliensis*-induced attenuation of arthritis. Untreated double *IL-4*/*IL-13* knockout mice displayed more severe arthritis than wild-type controls. In addition, the beneficial effect of the *N. brasiliensis* infection was not observed in the case of IL-4/13–deficient mice [156].

Administration of IL-4 in collagen-induced arthritis resulted in a delay of arthritis onset and disease amelioration [157,158,159]. Even more spectacular results were observed in experiments with PGIA. Implementation of IL-4 treatment in this arthritis model was associated with complete inhibition of the disease [160]. Mice deficient for the *IL-4* gene were also investigated in another study using PGIA model [131]. Following proteoglycan immunization, mice lacking *IL-4* exhibited increased risk of RA development, enhanced disease severity, and higher levels of proinflammatory cytokines, such as IL-1β, IL-6, IL-12, TNF, and INF, than wild-type controls [161]. Furthermore, in the study by Cao et al., mice with selective depletion of IL-4R in macrophages and neutrophils were examined using a PGIA model of arthritis [132]. Macrophage/neutrophil-specific IL-4R mice displayed significant exacerbation of both arthritis onset and severity, implying a crucial role of IL-4 in macrophages and neutrophils activity [162]. In addition, DCs genetically modified to express IL-4 were examined in the CIA model. Administration of IL-4–transduced DCs resulted in an inhibition of severity and disease onset [163]. Further experiments with genetically engineered DCs revealed that IL-4-transduced DCs induce primary Th2 immune responses and inhibit IL-17 production by T cells [164]. Additionally, the *IL-4* gene therapy approach in AIA resulted both in vitro and in vivo in an inhibition of angiogenesis and decreased neovascularization [165].

A recent study by Haikal et al. documented that impressive disease-amelioration effect in the CIA model was obtained after application of combination therapy of IL-4 with mesenchymal stromal cells [166]. Furthermore, the effect of IL-4 neutralization on oral administration of the inducing antigen in AIA was studied by Yoshino et al. Oral administration of the pathogenic antigen resulted in disease suppression, and a treatment with monoclonal antibody against IL-4 abrogated this beneficial effect [167]. A protective capacity of IL-4 against metalloproteinase-induced cartilage damage was also reported in an experimental ICA model [168]. On the other hand, some studies have demonstrated conflicting results. In an autoantibody-mediated model of RA with K/BxN mice, Ohmura et al. observed reduced symptoms of arthritis in mice deficient in IL-4 as compared to mice expressing this cytokine [169]. In addition, Svensson et al., in experiments with a new variant of the CIA model, reported that IL-4-deficient mice exhibited diminished RA severity and less frequently developed the disease [170].

An interesting study assessing impact of IL-4 on human ST engrafted in severe combined immunodeficient (SCID) mice was performed by Jorgensen et al. In this study, SCID mice were engrafted with human rheumatoid synovium and cartilage and treated thereafter with IL-4 injections. Cartilage degradation was found to decrease in IL-4 treated mice, what suggests a chondroprotective effect of IL-4 [171].

In conclusion, in experimental models of RA (both CIA and AIA), IL-4/IL-13 administration resulted in inhibition of levels of proinflammatory cytokines and angiogenesis, reduction of joint inflammation and disease amelioration. On the other hand, *IL-4* gene depletion was associated with increased levels of proinflammatory cytokines, enhanced severity and disease exacerbation.

## 11. IL-4/IL-13 in Human Studies

Human studies exploring the therapeutic potential of IL-4 and/or Il-13 cytokines in arthritis are required, as a boosting of anti-inflammatory cytokines seems to be a very promising approach in arthritis. At present, data regarding IL-4 and IL-13 administration are very limited. To date, IL-4 therapy has mostly been investigated as an anti-cancer agent in a series of phase I/II clinical trials. A recombinant human IL-4 (rhuIL-4) was administered to patients diagnosed with various neoplastic disorders, such as Hodgkin’s and non-Hodgkin’s lymphoma, malignant melanoma, renal cell carcinoma, pancreatic adenocarcinoma, non-small cell lung cancer, and chronic lymphocytic leukemia [172,173,174,175,176]. Various dose ranges, schedules, and routes of administration of rhuIL-4 were assessed in these studies. Although the anticancer potential of this cytokine was relatively modest and the results were disappointing, the IL-4 treatment regimen was well tolerated.

With respect to autoimmune diseases, the rhuIL-4 therapeutic potential was only investigated in patients diagnosed with severe psoriasis [177]. The study conducted by Ghoreshi et al. in 2013 yielded very promising results documenting that psoriasis patients administered with rhuIL-4 exhibited significant improvement following six weeks of IL-4 therapy. In this study, rhuIL-4 was administered subcutaneously, in doses ranging from 200 to 500 ng/kg of body weight, three times a day, and was well tolerated. This study constitutes strong evidence supporting the hypothesis that IL-4-based therapy can be clinically effective in autoimmune disorders. The promising outcome of IL-4 therapy in psoriasis implies an importance of evaluation of IL-4 treatment in other autoimmune diseases, such as RA [48,178,179]. Future clinical studies should assess both efficacy and safety of IL-4 and Il-13 cytokines administration in inflammatory arthritis, either in monotherapy or in combination therapy with other drugs. It is also of interest to explore combined potential of these two cytokines as an IL-4-IL-13 fusion protein. The evaluation of most optimized dose ranges, routes of administration, and application schedules should be also performed to allow to provide a maximal efficacy with minimum side effects.

## 12. Therapeutic Perspectives

As mentioned previously, clinical trials involving rhuIL-4 performed to date have evidenced its safety and good tolerability. These results suggest that the IL-4 protein could potentially be employed in treatment of patients diagnosed with autoimmune diseases, such as inflammatory arthritis. However, cytokine-based therapies present considerable challenges, including short half-life, poor bioavailability, inadequate efficacy, and dose-limiting toxicities [180]. Recent advances in this field offer various strategies to increase the therapeutic potential and minimize side effects of administered cytokines. An improvement in IL-4 therapeutic effect and development of a therapeutically efficient IL-13 protein might be obtained by employing molecular engineering techniques [181]. Promising strategies include a generation of new cytokine mutants (superkines) or cytokine–antibody fusion constructs (immunokines) displaying increased stability and specificity, prolonged half-life, localized delivery, and enhanced therapeutic efficacy [182].

Recently, two engineered IL-4 superkines have been generated [183]. Additionally, development of the IL-4-IL-13 fusion protein, combining biologic activity of these two cytokines, might unveil a promising immunotherapeutic agent. These two cytokines combined into one molecule might be more effective than stand-alone cytokines administered separately or in a combination therapy. Recently, a publication by Eijkelkamp documented the development of a IL4-10 fusion protein and in vivo study of this protein using two mouse models of persistent inflammatory pain [184]. An intrathecal administration of the IL4-10 fusion protein was very efficacious and induced complete resolution of inflammatory pain. The IL4-10 fusion protein was further examined in in vitro as well as in vivo models of osteoarthritis and chondroprotective, anti-inflammatory, and analgesic activity of this protein was reported [185].

Another potential approach is to employ gene therapy using viral or non-viral vectors to deliver gene encoding protein of interest [186]. The microencapsulation technique represents another potential strategy for a successful administration of IL-4/IL-13 [187]. A recent study demonstrated that IL-4/IL-13-encapsulated biodegradable microspheres displayed prolonged half-life and effectively suppressed inflammatory activity of osteoarthritic chondrocytes in in vitro assay [188].

## 13. Conclusions

IL-4 and IL-13 are critical components of Th-2-mediated immunity, and they play a crucial role in pathogenesis of allergic inflammation. They are secreted by various cell types and act through STAT6 signaling pathway. They exert effect on a wide range of cells, such as B cells, eosinophils, basophils, monocytes, and fibroblasts. Blocking IL4/IL-13 yielded promising results in Th-2-induced diseases, such as asthma or atopic dermatitis. However, accumulating evidence implies their role in a pathogenesis of autoimmune-mediated diseases such as inflammatory arthritis. Results obtained from animal, as well as ex vivo, models of arthritis suggest that the IL-4/IL-13 anti-inflammatory properties might be beneficial in the context of inflammatory arthritis treatment. Induction of the IL-4/IL-13 signaling pathway might constitute novel powerful approach in therapy of inflammatory arthritis. Therefore, further studies are crucial to uncover and understand the mechanism of IL-4/IL-13 action and their exact role in inflammatory arthritis.

## Figures and Tables

**Figure 1 cells-10-03000-f001:**
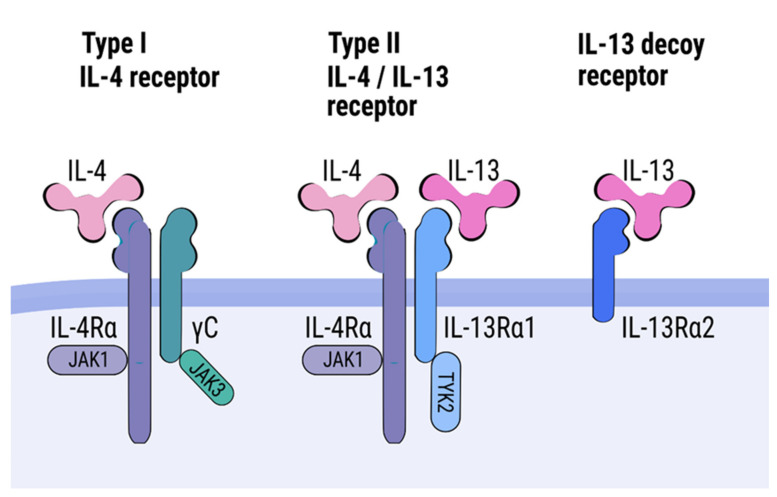
IL-4 receptor alpha (IL-4Rα) constitutes subunits of two heterodimeric receptors, named type I and type II receptors. The type I receptor, composed of IL-4Rα and common cytokine receptor γ-chain (γc), interacts only with IL-4. The type II receptor is formed from IL-4Rα and IL-13Rα1 subunits and interacts with either IL-13 or IL-4. IL-13 also displays the capacity to bind IL-13Rα2, which is regarded as a decoy receptor. A binding of a ligand by type I and II receptors results in an activation of Janus family kinases (JAK1, JAK2, and JAK3) followed by phosphorylation of a signal transducer and activator transcription 6 (STAT6).

**Figure 2 cells-10-03000-f002:**
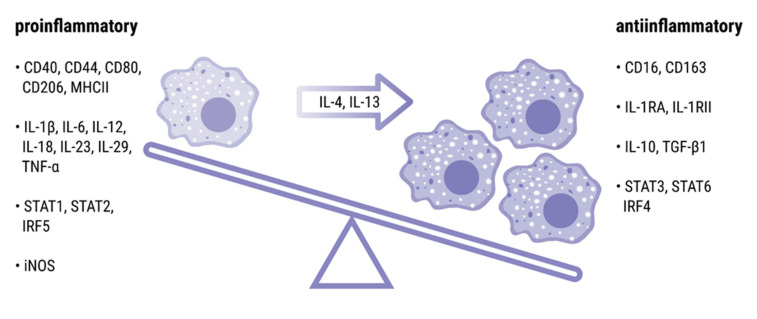
Macrophages’ polarization upon the action of IL-4 and IL-13 from classically activated/inflammatory (M1) phenotype to alternatively activated (M2) phenotype.

**Figure 3 cells-10-03000-f003:**
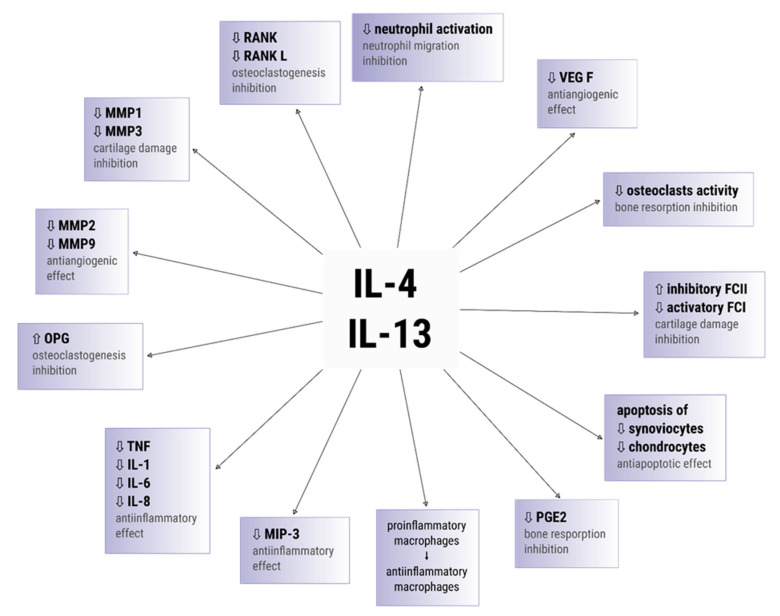
IL-4/IL-13 functions in inflammatory arthritis.

## Data Availability

Not applicable.

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
