# Peer review of "Significance of Interleukin (IL)-4 and IL-13 in Inflammatory Arthritis"

_cells, 2021, doi:10.3390/cells10113000_

Round 1

Reviewer 1 Report

Referee’s comment

Article n° cells-1395489

Title: Significance of interleukin (IL)-4 and IL-13 in inflammatory 2 arthritis

Authors:  Iwaszko M,  BiaÅ‚y S,  Bogunia-Kubik K

General comment

This is an interesting and current review which informs us about the progress made in terms of modulating inflammation. The aims and objectives are clear, the paragraphs are well structured and the data are correctly presented. The paper should be of interest to scientist working in the field of arthritis and osteoarthritis as well as others with closely related research interest. Therefore, this article contain many novelty  of interest for the research community and  in conclusion, considering the above matters, I retain that this manuscript could be accepted for publication in  Cells. However, I have only some minor suggestions to make the paper more suitable:

1) Do CD44, IL-1beta, IL-29 and other cytokines  participate in the activation of macrophages? The list in Fig. 1 should be updated. Or, if the authors have different data, at least explain the reasons in the text why these cytokines do not activate macrophages.

2) A cartoon showing the involvement of Il-4 and IL-13 in arthritis could help the reader to better comprise the action exerted.

3) A short paragraph relating to possible human studies (in vivo / in vitro) should be included in the text.

4) A separate paragraph reporting the future prospects and strategies should be inserted before conclusions

Author Response

The authors would like to thank the Referee for critical review of the manuscript. The authors appreciate valuable remarks of the Reviewer that led to improved quality of the manuscript. All the comments were taken into consideration and the manuscript has been enhanced according to the Reviewer’s suggestions.

1) Do CD44, IL-1beta, IL-29 and other cytokines  participate in the activation of macrophages? The list in Fig. 1 should be updated. Or, if the authors have different data, at least explain the reasons in the text why these cytokines do not activate macrophages.

RESPONSE: According to the Reviewer's suggestion, the Fig.1 has been improved.

2) A cartoon showing the involvement of Il-4 and IL-13 in arthritis could help the reader to better comprise the action exerted.

RESPONSE: We fully agree with the Reviewer and an additional figure (Figure 3) has been provided in the revised version of the manusript.

3) A short paragraph relating to possible human studies (in vivo / in vitro) should be included in the text.

RESPONSE: As the Reviewer suggested, a paragraph related to human studies has been included in the revised version of the manusript.

4) A separate paragraph reporting the future prospects and strategies should be inserted before conclusions

RESPONSE: As advised by the Reviewer, an additional paragraph regarding future prospects and strategies has been provided in the revised version of the manusript.  

Reviewer 2 Report

Often the available data about the significance of IL-4 and IL-13 in inflammatory arthritis are contradictory. Therefore, it would be helpful to add a short sentence at the end of each paragraph to summarize the most important finding. This would facilitate understanding the manuscript.  I recommend proofreading of the manuscript.

Author Response

The authors would like to thank the Referee for critical review of the manuscript. The authors appreciate valuable remarks of the Reviewer that led to improved quality of the manuscript. All the comments were taken into consideration and the manuscript has been enhanced according to the Reviewer’s suggestions.

According to the Reviewer's suggestion, conclusive sentences at the end of most paragraphs have been provided. In addition, Fig.3 summarizing IL-4/IL-13 functions has been added in revised version of the manuscript.

The manuscript was rereaded and corrected.

Reviewer 3 Report

Iwaszko et al have summarized the information on the role of IL-4 and IL-13 in some inflammatory arthritis with special emphasis on their involvement in rheumatoid arthritis. In addition to describing both cytokine polymorphisms in these pathologies, this review includes the existing data to date on their expression levels and their regulatory action on several key cells and molecules involved in rheumatoid arthritis pathogenesis.

I hope the following comments will help them to improve the manuscript.

  1. This is a correctly structured review paper in which a logical organization of the content can be appreciated, which helps a better understanding of the information. However, some paragraphs of the text have been included in a location that would not correspond to its content. For example, the third paragraph on page 3 (lines 95-101) talks about serum and synovial fluid levels of IL-13. The information in this paragraph should be placed in section 3 (line 147) dedicated to " IL-4 and IL-13 serum concentrations in inflammatory arthritis". The same applies to lines 337-339: they include information related to the CIA model together with results obtained in the SIA model. It would be recommendable to present all the information related to each animal model together (the data of the CIA model are presented in the first paragraph of the section (lines 304-315).
  2. The paragraph describing IL-4 and IL-13 receptors (lines 54-70) is very confusing. The wording should be revised to facilitate understanding. In fact, this same information is included in the figure caption 1 and is easier to read.
  3. Abbreviations must be specified the first time they appear in the text. Multiple abbreviations are used throughout the text with no previous explanation of their meaning (some examples: line 85 “ESR”; line 107 “DAS28”; line 124 “RF”). Also, conversely, the complete name of abbreviations that had already been specified in the text is used (i.e. line 159 nd line 173 “PB”). There are even descriptions of abbreviations that are specified twice (some examples: lines 215 and 353 “DC”, lines 358 and 364 “AIA”). It would be advisable, therefore, to carefully check the abbreviations used in the text in order to use them appropriately.
  4. Line 221: Please correct “…FccRIIb…”
  5. Line 232: Please remove the brackets from “(MIP)-3”
  6. In section 4 regarding the effect of IL-4 and IlL13 on macrophage polarization, no reference is made to the role of M1 or M2 in the pathogenesis of inflammatory arthritis. The modulatory effects of both cytokines on macrophages should be contextualized in the pathologies studied.
  7. Line 290: Please correct “…nuclear factor-/span>B…”
  8. In section 6, the same cell type is referred to by different names, which makes it difficult to understand. Fibroblasts are mentioned in lines 240, 253 and 254 under different terms. Only in line 254 it is specified that they are fibroblast-like synoviocytes and even its abbreviation is indicated.

Author Response

The authors would like to thank the Referee for critical review of the manuscript. The authors appreciate valuable remarks of the Reviewer that led to improved quality of the manuscript. All the comments were taken into consideration and the manuscript has been enhanced according to the Reviewer’s suggestions.

  1. This is a correctly structured review paper in which a logical organization of the content can be appreciated, which helps a better understanding of the information. However, some paragraphs of the text have been included in a location that would not correspond to its content. For example, the third paragraph on page 3 (lines 95-101) talks about serum and synovial fluid levels of IL-13. The information in this paragraph should be placed in section 3 (line 147) dedicated to " IL-4 and IL-13 serum concentrations in inflammatory arthritis". The same applies to lines 337-339: they include information related to the CIA model together with results obtained in the SIA model. It would be recommendable to present all the information related to each animal model together (the data of the CIA model are presented in the first paragraph of the section (lines 304-315).

RESPONSE: We fully agree with the Reviewer and these issues have been corrected.

  1. The paragraph describing IL-4 and IL-13 receptors (lines 54-70) is very confusing. The wording should be revised to facilitate understanding. In fact, this same information is included in the figure caption 1 and is easier to read.

RESPONSE: According to the Reviewer's suggestion, this paragraph has been shortened and improved.

  1. Abbreviations must be specified the first time they appear in the text. Multiple abbreviations are used throughout the text with no previous explanation of their meaning (some examples: line 85 “ESR”; line 107 “DAS28”; line 124 “RF”). Also, conversely, the complete name of abbreviations that had already been specified in the text is used (i.e. line 159 nd line 173 “PB”). There are even descriptions of abbreviations that are specified twice (some examples: lines 215 and 353 “DC”, lines 358 and 364 “AIA”). It would be advisable, therefore, to carefully check the abbreviations used in the text in order to use them appropriately.

RESPONSE: We fully agree with the Reviewer, we reread the manuscript and provided appropriate corrections.

  1. Line 221: Please correct “…FccRIIb…”

RESPONSE: This point has been corrected.

  1. Line 232: Please remove the brackets from “(MIP)-3”
    RESPONSE: This point has been corrected.

  2. In section 4 regarding the effect of IL-4 and IlL13 on macrophage polarization, no reference is made to the role of M1 or M2 in the pathogenesis of inflammatory arthritis. The modulatory effects of both cytokines on macrophages should be contextualized in the pathologies studied.

RESPONSE: We fully agree with the Reviewer and additional references have been provided.

  1. Line 290: Please correct “…nuclear factor-/span>B…”

RESPONSE: This has been corrected.

  1. In section 6, the same cell type is referred to by different names, which makes it difficult to understand. Fibroblasts are mentioned in lines 240, 253 and 254 under different terms. Only in line 254 it is specified that they are fibroblast-like synoviocytes and even its abbreviation is indicated.

RESPONSE: Thank You for this remark, it has been corrected.

Reviewer 4 Report

Cells-1395489

Significance of interleukin (IL)-4 and IL-13 in inflammatory arthritis

This review article summarizes the role of the cytokines IL-4 and IL-13 in the pathogenesis of inflammatory arthritides. In the manuscript, (I) the basic features/characteristics of IL-4/-13, the interaction with the respective receptors, and the subsequently activated signaling pathways are described, (II) the influence of IL-4/-13 polymorphisms and IL-4/-13 levels on inflammation (including macrophage polarization and pro-inflammatory cytokine production) in different arthritides is discussed, and (III) data from relevant (disease) models are mentioned (including in vitro, ex vivo, and animal models).

The review article adequately covers the selected topic, reflects relevant parts of both basic reports and the latest literature, and provides an informative overview for the reader. However, some critical points/aspects have to be addressed to further improve the quality of the manuscript.

  1. The text of the manuscript has to be revised (some sentences with confusing syntax and phrasing; inconsistent style (e.g., concerning the description of polymorphisms); comma setting; in part, singular and plural are mixed up; use superscript symbols where appropriate).
  2. I assume that at the end of line 8 (“… along with IL-4 …”) IL-3 is meant (instead of a second mention of IL-4).
  3. All abbreviations have to be defined in the text.
  4. In the Introduction, please better point out the role of IL-4/-13 as anti-inflammatory cytokines.
  5. Please include a short paragraph on IL-4/-13-inducing signals/stimuli and the major signaling pathways/transcription factors regulating IL-4/-13 expression.
  6. Please also provide more basic information on IL-4 and IL-13 genes/proteins (length, transcriptional/translational regulation, mode of secretion, …) .
  7. Please consistently provide the position/location, the major/minor allele, and the rs-number for all polymorphisms mentioned. Where appropriate, please also include the information whether the reported effects refer to the homo- or the heterozygous occurrence of the respective polymorphisms. For aa exchanges (e.g., in the cytokine receptors), please define the affected protein domain.
  8. For paragraphs describing large amounts of data (and especially for paragraphs providing contradictory literature results), a few concluding/classifying sentences that interpret the reported findings in the context of inflammatory arthritis pathogenesis would facilitate the understanding of the text.
  9. Introduction, lines 54-70: This paragraph appears quite unstructured. Please improve.
  10. Figure 1: Revise the Figure to better point out that type II is only one receptor.
  11. Chapter 2, lines 103-115: Is the polymorphism located at position -590 or 590? Please clarify.
  12. Chapter 3, lines 148-178: To me, this section raises the question whether there might be a time-dependent effect (i.e., activating influence of IL-4/-13 on RA in the onset of RA/during early RA and an inhibiting influence on late RA). Please comment on that.
  13. Chapter 4, lines 180-190: Please comment on the influence of IL-4/-13 on surface marker expression (as indicated in Figure 2).
  14. Chapter 7, line 279: Which agent(s) was/were used for cell stimulation (in the presence of mtx)?
  15. In the reference List, reference 83 is missing.

Author Response

The authors would like to thank the Referee for critical review of the manuscript. The authors appreciate valuable remarks of the Reviewer that led to improved quality of the manuscript. All the comments were taken into consideration and the manuscript has been enhanced according to the Reviewer’s suggestions.

1. The text of the manuscript has to be revised (some sentences with confusing syntax and phrasing; inconsistent style (e.g., concerning the description of polymorphisms); comma setting; in part, singular and plural are mixed up; use superscript symbols where appropriate).

RESPONSE: We fully agree with the Reviewer, we reread the manuscript and provided appropriate corrections.

2. I assume that at the end of line 8 (“… along with IL-4 …”) IL-3 is meant (instead of a second mention of IL-4).

RESPONSE: Yes, we made a mistake and correction was provided.

3. All abbreviations have to be defined in the text.

RESPONSE: We fully agree with the Reviewer and this issue has been corrected.

4. In the Introduction, please better point out the role of IL-4/-13 as anti-inflammatory cytokines.

RESPONSE: According to the Reviewer's suggestion, additional sentences regarding IL-4/-13 anti-inflammatory properties have been provided in the Introduction section.

5. Please include a short paragraph on IL-4/-13-inducing signals/stimuli and the major signaling pathways/transcription factors regulating IL-4/-13 expression.

RESPONSE: As the Reviewer suggested, a paragraph describing regulation of IL-4/-13 expression has been included in revised version of the manuscript.

6. Please also provide more basic information on IL-4 and IL-13 genes/proteins (length, transcriptional/translational regulation, mode of secretion, …) .

RESPONSE: In agreement with the Reviewer's advice these informations have been added in the Introduction section.

7. Please consistently provide the position/location, the major/minor allele, and the rs-number for all polymorphisms mentioned. Where appropriate, please also include the information whether the reported effects refer to the homo- or the heterozygous occurrence of the respective polymorphisms. For aa exchanges (e.g., in the cytokine receptors), please define the affected protein domain.

RESPONSE: According to the Reviewer's suggestion, these informations have been provided in revised version of the manuscript.

8. For paragraphs describing large amounts of data (and especially for paragraphs providing contradictory literature results), a few concluding/classifying sentences that interpret the reported findings in the context of inflammatory arthritis pathogenesis would facilitate the understanding of the text.

RESPONSE: We fully agree with the Reviewer and conclusive sentences at the end of most paragraphs have been provided.

9. Introduction, lines 54-70: This paragraph appears quite unstructured. Please improve.

RESPONSE: According to the Reviewer's suggestion, this paragraph has been shortened and improved.

10. Figure 1: Revise the Figure to better point out that type II is only one receptor.

RESPONSE: As the Reviewer suggested, Figure 2 has been improved.

11. Chapter 2, lines 103-115: Is the polymorphism located at position -590 or 590? Please clarify.

RESPONSE: It has been clarified.

12. Chapter 3, lines 148-178: To me, this section raises the question whether there might be a time-dependent effect (i.e., activating influence of IL-4/-13 on RA in the onset of RA/during early RA and an inhibiting influence on late RA). Please comment on that.

RESPONSE: Thank You for this remark, this aspect has been discussed in revised version of the manuscript.

13. Chapter 4, lines 180-190: Please comment on the influence of IL-4/-13 on surface marker expression (as indicated in Figure 2).

RESPONSE: We fully agree with the Reviewer and additional sentences have been provided in this chapter.

14. Chapter 7, line 279: Which agent(s) was/were used for cell stimulation (in the presence of mtx)?

RESPONSE: This information has been provided in this chapter.

15. In the reference List, reference 83 is missing.

RESPONSE: Thank You, this point has been corrected.

Round 2

Reviewer 4 Report

Cells-1395489

Significance of interleukin (IL)-4 and IL-13 in inflammatory arthritis

The manuscript provides a revised version of the manuscript “Significance of interleukin (IL)-4 and IL-13 in inflammatory arthritis”. The manuscript has been improved considerably and the reviewers’ comments have been adequately addressed.